

# Interpretation of exercise-induced changes in human skeletal muscle mRNA expression depends on the timing of the post-exercise biopsies

Jujiao Kuang[1,2], Cian McGinley[3], Matthew J-C Lee[1],
Nicholas J. Saner[1,4], Andrew Garnham[1] and David J. Bishop[1]

[1] Institute for Health and Sport, Victoria University, Melbourne, Victoria, Australia
[2] Australia Institute for Musculoskeletal Sciences, Melbourne, Victoria, Australia
[3] Sportscotland Institute of Sport, Stirling, United Kingdom
[4] Human Integrative Physiology, Baker Heart and Diabetes Institute, Melbourne, Victoria, Australia

## ABSTRACT

**Background:** Exercise elicits a range of adaptive responses in skeletal muscle, which include changes in mRNA expression. To better understand the health benefits of exercise training, it is important to investigate the underlying molecular mechanisms of skeletal muscle adaptation to exercise. However, most studies have assessed the molecular events at only a few time-points within a short time frame post-exercise, and the variations of gene expression kinetics have not been addressed systematically.
**Methods:** We assessed the mRNA expression of 23 gene isoforms implicated in the adaptive response to exercise at six time-points (0, 3, 9, 24, 48, and 72 h post exercise) over a 3-day period following a single session of high-intensity interval exercise.
**Results:** The temporal patterns of target gene expression were highly variable and the expression of mRNA transcripts detected was largely dependent on the timing of muscle sampling. The largest fold change in mRNA expression of each tested target gene was observed between 3 and 72 h post-exercise.
**Discussion and Conclusions:** Our findings highlight an important gap in knowledge regarding the molecular response to exercise, where the use of limited time-points within a short period post-exercise has led to an incomplete understanding of the molecular response to exercise. Muscle sampling timing for individual studies needs to be carefully chosen based on existing literature and preliminary analysis of the molecular targets of interest. We propose that a comprehensive time-course analysis on the exercise-induced transcriptional response in humans will significantly benefit the field of exercise molecular biology.

Corresponding author
David J. Bishop,
David.Bishop@vu.edu.au

## INTRODUCTION

Exercise is a powerful stimulus affecting skeletal muscle, leading to improvements in cardiovascular function, mitochondrial content and function, and whole-body metabolism (*Bishop et al., 2019*; *Cornelissen & Smart, 2013*; *Granata et al., 2021*; *Lavie et al., 2015*;

*Philippou et al., 2019*). The molecular basis of skeletal muscle adaptations to exercise fundamentally involve modified protein content and enzyme activity, mediated by an array of pre- and post-transcriptional processes, as well as translational and post-translational control (*Egan, Hawley & Zierath, 2016*; *Egan & Zierath, 2013*). From the onset of exercise, muscle contraction can induce disruptions to muscle homeostasis, including mechanical stress, calcium release, ATP turnover, changes to mitochondrial redox state, and reactive oxygen species (ROS) production. These cellular perturbations lead to the activation of signaling molecules that activate a range of transcription factors and coactivators, such as peroxisome proliferator-activated receptor gamma coactivator 1α (PGC-1α) and p53 (*Egan & Zierath, 2013*). In turn, changes in these and other proteins help to coordinate the transcription of genes associated with mitochondrial biogenesis, fat metabolism, and glucose metabolism (*Egan & Zierath, 2013*).

It has been proposed that training-induced adaptions are due to the cumulative effect of each single exercise session, and that investigating exercise-induced changes in mRNA after a single exercise session can provide important information about the likely adaptations to repeated exercise sessions (*i.e.*, exercise training) (*Egan et al., 2013*; *Perry et al., 2010*). Using both quantitative real-time PCR (qPCR) and whole-transcriptome analysis, many studies have provided a better understanding of the transcriptional response to exercise. However, relatively few studies have sampled muscle at multiple times post exercise or taken biopsies beyond the first 24 h (*Andrade-Souza et al., 2020*; *Catoire et al., 2014*; *Gidlund et al., 2015*; *Granata et al., 2020*; *Hyldahl et al., 2015*; *Jensen et al., 2015*; *Lindholm et al., 2014*; *Liu et al., 2010*; *Mahoney et al., 2005*; *McLean et al., 2015*; *Murton et al., 2014*; *Neubauer et al., 2014*; *Ogborn et al., 2015*; *Pillon et al., 2020*; *Raue et al., 2012*; *Rowlands et al., 2011*; *Thalacker-Mercer et al., 2013*; *Vissing & Schjerling, 2014*). A recent study has employed a meta-analysis to profile the skeletal muscle transcriptome using 66 published datasets, providing a useful resource to check the expression of genes of interest in response to a single session of exercise, exercise training, or inactivity (*Pillon et al., 2020*). However, many of these findings strongly rely on the timing of the post-exercise biopsy used in individual studies, and some of these studies provide limited information about the transient nature of exercise-induced changes in gene expression due to few sampling time-points post-exercise. Thus, while there is a general understanding that exercise-induced changes in mRNA expression are time-dependent, more studies are required that extend these analyses to multiple time-points sampled over a prolonged post-exercise period.

The absence of a strong justification for the choice of post-exercise biopsy time-point has important implications for our understanding of molecular adaptations to exercise. For example, while *Scribbans et al. (2017)* reported there was not a systematic upregulation of nuclear and mitochondrial genes 3 h post-exercise, they also noted that their chosen biopsy time-point might have failed to capture any changes that occurred later in the post-exercise period. Similarly, the lack of increase in *p53* mRNA after exercise has been interpreted as evidence that post-translational modification is more important in regulating protein levels of p53 (*Saleem & Hood, 2013*); but, it is possible exercise-induced changes in *p53* mRNA have been missed by the biopsy time-points chosen to date. Thus, it
is clear that the choice of post-exercise biopsy time-point can influence the interpretation of the transcriptional response to exercise (*McGinley & Bishop, 2016a*; *Yang et al., 2005*).

The purpose of this research was to investigate the temporal expression of commonly assessed, exercise-responsive genes after a single session of endurance-based exercise. We assessed the mRNA expression of key transcription factors associated with exercise-induced mitochondrial biogenesis (PGC-1α and p53), as well as other genes that have potential roles in mitochondrial and metabolic adaptions to exercise. We hypothesized that different genes would elicit different temporal patterns of expression. The results have helped to highlight the importance of appropriate muscle sample timing and to provide recommendations for designing future studies examining molecular responses to exercise.

## MATERIALS AND METHODS

### Participants

As part of a larger project (*McGinley & Bishop, 2016a*; *McGinley & Bishop, 2016b*), 16 recreationally-active men were fully briefed on the procedures, risks, and benefits associated with participating, before providing written informed consent to participate and for their data to be used in the present study. However, due to the availability of muscle samples, data from only nine participants were available for the present study (mean (SD); age: 22 (4) y; height: 179.5 (7.9) cm; mass: 81.4 (14.3) kg; $\dot{V}O_{2peak}$: 3.9 (0.3) L•min$^{-1}$ at baseline). All procedures were approved by the Victoria University Human Research Ethics Committee, and the ethics number is HRETH 11/289.

### Experimental design

Following a familiarisation trial (performed on a separate day; mean (SD), 3 (8) days prior to resting muscle biopsy at baseline), participants completed a graded-incremental exercise test (GXT) to determine baseline levels of peak aerobic power ($\dot{W}_{peak}$) and the power at the first lactate threshold ($\dot{W}_{LT}$). This information was used to individualize the intensity of a single session of high-intensity interval exercise (HIIE). The resting muscle biopsy at baseline was taken on a day before the GXT except for one participant. As part of another study (*McGinley & Bishop, 2016b*), the participants completed a 4 week high-intensity interval training (HIIT) intervention, training 3 days a week (12 sessions in total). The HIIE session in the present study was the final HIIE session of the HIIT intervention, which was 2 to 4 days after the penultimate HIIE session. To minimise the number of muscle biopsies for each participant, a resting muscle biopsy was taken before starting the 4-week training period (Week 0) and used as the baseline value, as done in previous research (*Mahoney et al., 2005*; *Neubauer et al., 2014*). Previously published evidence has reported that the expression of the majority of genes (24,686 out of 24,838 genes assessed) is not changed at rest before and after 6 weeks of exercise training (*Miyamoto-Mikami et al., 2018*). Therefore, it is unlikely the fold change in gene expression changes measured in the present study are due to a training effect. Six more biopsies were sampled following the single HIIE session immediately (0 h), and at 3, 9, 24, 48, and 72 h post-exercise (Fig. 1).
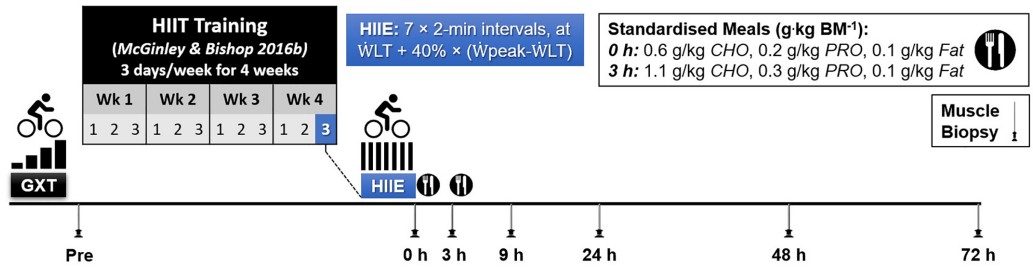

**Figure 1 Experimental Design.** Abbreviations: BM, body mass; CHO, carbohydrate; PRO, protein; GXT, graded exercise test; HIIE, high-intensity interval exercise; HIIT, high-intensity interval training; $\dot{W}_{LT}$, power at the first lactate threshold; and $\dot{W}_{peak}$, peak aerobic power.

## Graded exercise test

The GXT was conducted pre-training to determine the $\dot{W}_{LT}$ and $\dot{W}_{peak}$. All trials were conducted in the morning (6.30–11.30). To determine the $\dot{W}_{LT}$, a 20-gauge intravenous cannula was inserted into an antecubital vein; venous blood was sampled at rest and at the end of every stage of the GXT, as previously described (*McGinley & Bishop, 2016b*). The mean coefficient of variation (CV) for duplicate blood lactate measurements was 4.6%. The $\dot{W}_{LT}$ was identified as the power at which venous blood lactate increased 1 mM above baseline, and was calculated using Lactate-E software (*Newell et al., 2007*).

The GXT was performed on an electromagnetically-braked cycle ergometer (Excalibur Sport, Lode, Groningen, The Netherlands), using an intermittent protocol consisting of 4-min exercise stages separated by 30-s of rest (*Jamnick et al., 2018*). The initial load (90 to 150 W) was ascertained during the familiarisation GXT, and subsequently increased by 30 W every 4.5 min, with the aim of minimizing the total number of stages to a maximum of 10. Participants were required to maintain a cadence of 70 rpm, and consistent verbal encouragement was provided during the latter stages. The test was terminated either volitionally by the participant, or by the assessors if the participant could no longer maintain the required cadence (±10 rpm) despite strong verbal encouragement. $\dot{W}_{peak}$ was calculated as previously reported (*Kuipers et al., 1985*):

$$\dot{W}_{peak} = \dot{W}_{final} + \left( \frac{t}{240} \times 30 \right)$$

where $\dot{W}_{final}$ was the power output of the last completed stage and $t$ was the time in seconds of any final incomplete stage.

## High-intensity interval exercise

Following an overnight fast, the participants performed a single session of high-intensity interval exercise (HIIE), between 6.30 and 8.00. The exercise consisted of 7 2-min intervals performed on an electromagnetically-braked cycle ergometer (Velotron, Racer-Mate, Seattle, WA, USA), separated by 1 min of passive recovery (2:1 work:rest). A standardized 5-min steady-state warm-up at 75 W was completed beforehand. The exercise intensity was set to $\dot{W}_{LT}$ plus 40% of the difference between $\dot{W}_{LT}$ and $\dot{W}_{peak}$; *i.e.*, ($\dot{W}_{LT}$) + (40 %)

$(\dot{W}_{peak} - \dot{W}_{LT})$. Power at the LT was 65 (6)% of $\dot{W}_{peak}$, resulting in the HIIE session being undertaken at 79 (4)% of $\dot{W}_{peak}$ (mean (SD)).

## Dietary control

Participants performed the HIIE session following an overnight fast. Participants were requested to refrain from caffeine consumption on the day of the HIIE, to not ingest any dietary supplements, and to abstain from both alcohol consumption and exercise in the preceding 24-h period. Following the final HIIE session, participants were provided with two meals, totaling one-third of their daily energy requirement, based on their predicted basal metabolism (*Harris & Benedict, 1918*), and allowing for a 1.4 activity correction factor (*Durnin, 1996*).

Following the 0 h biopsy (*i.e.*, immediately post-HIIE), participants received a standardized breakfast (1,416 (230) kJ; mean (SD)), with a target macronutrient content (expressed in g per kg BM) of: 0.6 g/kg carbohydrate, 0.2 g/kg protein, and 0.1 g/kg fat. Following the 3 h biopsy, participants received a standardized meal (2,456 (399) kJ), consisting of: 1.1 g/kg carbohydrate, 0.3 g/kg protein and 0.1 g/kg fat. The total relative macronutrient intake was therefore 64% carbohydrate, 16% protein, and 20% fat. Excluding the standardized meals provided, participants were instructed to ingest only water *ad libitum* until after the 9-h biopsy. Light activities (*e.g.*, walking) were permitted between the 0- and 9-h biopsies. All other biopsies (week 0, plus 24, 48, and 72 h) were sampled in the morning following an overnight fast, with participants refraining from additional exercise or alcohol consumption until after the final muscle biopsy.

## Muscle sampling

Muscle biopsies were performed by a qualified medical doctor and taken from the non-dominant leg at rest pre-HIIT (Week 0) and 4 weeks later after the final HIIE session–immediately (0 h), 3, 9, 24, 48, and 72 h post-exercise. Muscle samples were taken from the vastus lateralis approximately one third of the distance between knee and hip. Subsequent samples were taken approximately 1 cm proximal to the previous biopsy site. Local anesthesia (Lidocaine, 1%) was injected into biopsy site and a 5-mm incision was made once numb, and muscle was sampled using a suction-modified Bergström needle (*Bergström, 1962*). Samples were cleaned of excess blood, fat, and connective tissue before being immediately snap-frozen in liquid nitrogen, and then stored at −80 °C for subsequent analyses.

## RNA extraction

qPCR was performed on $n = 9$ at all time-points, except for at 72 h ($n = 6$), using methods established by our group (*Kuang et al., 2018*). An RNeasy Plus Universal Mini Kit was used to extract total RNA from approximately 20 mg of frozen muscle. Samples were homogenized using a QIAzol lysis reagent and a TissueLyser II (Qiagen, Valencia, USA). The instructions for the kit were modified slightly to increase RNA yield by replacing ethanol with 2-propanol and storing samples at −20 °C for 2 h (*Kuang et al., 2018*). Purification of RNA samples was performed according to kit instructions using a genomic

DNA (gDNA) eliminator solution containing cetrimonium bromide. A BioPhotometer (Eppendorf AG, Hamburg, Germany), was used to determine both the concentration and purity of the RNA samples (based on the $A_{260}/A_{280}$ ratio). RNA integrity of all samples was measured using a Bio-Rad Experion microfluidic gel electrophoresis system (Experion RNA StdSens Analysis kit) and determined from the RNA quality indicator (RQI: 8.8 (0.5)). RNA was stored at −80 °C until reverse-transcription was performed.

### Reverse transcription

One µg of RNA, in a total reaction volume of 20 µL, was reverse-transcribed to cDNA using a Thermocycler (Bio-Rad, Hercules, CA, USA) and iScript RT Supermix (Bio-Rad, Hercules, CA, USA) as per the manufacturer's instructions. Priming was performed at 25 °C for 5 min and reverse transcription for 30 min at 42 °C. All samples, including 1 RT-negative control, were run on the same plate. cDNA was stored at −20 °C until subsequent analysis.

### qPCR

Relative mRNA expression was measured by qPCR (QuantStudio 7 Flex; Applied Biosystems, Foster City, USA). Primers were designed using Primer-BLAST (*Ye et al., 2012*) and purchased from Sigma-Aldrich (see Table 1 for primer details). All reactions were performed in duplicate on 384-well MicroAmp optical plates (Applied Biosystems, Foster City, USA) using an epMotion M5073 automated pipetting system (Eppendorf AG, Hamburg, Germany). A total reaction volume of 5 µL contained 2 µL of diluted cDNA template (10- to 160-fold dilution), 2.5 µL of mastermix (SsoAdvanced Universal SYBR Green Supermix, Bio-Rad, Hercules, CA, USA), and 0.3 µL of primers (5 µM or 15 µM). All qPCR assays were run for 10 min at 95 °C, followed by 40 cycles of 15 s at 95 °C and 60 s at 60 °C. The expression of each target gene was normalized to the geometric mean of expression of the three most stably expressed reference genes out of 6 potential ones being tested (TBP, PPIA, and B2M) (*Vandesompele et al., 2002*), and using the $2^{-\Delta\Delta Cq}$ method (*Schmittgen & Livak, 2008*).

### Statistics

RefFinder was utilized for the statistical analysis of reference genes (*Xie et al., 2012*). For change in mRNA expression, statistical analyses were performed on the $2^{-\Delta Cq}$ data, but the relative expression is reported ($2^{-\Delta\Delta Cq}$). Geometric means and geometric standard deviations (geometric mean (GSD)) are reported. A Mann-Whitney U test was used to compare the difference between post-exercise time-points and baseline values as the normality of the distribution calculated with the Shapiro-Wilk test was rejected, which was confirmed by normal probability plot (Q-Q plot). Differentially expressed gene targets post-exercise were first determined using *a posteriori* information fusion scheme that combines the biological relevance (fold change) and the statistical significance ($p$ value); significance was defined as Xiao value <0.05 (*Deshmukh et al., 2021*; *Xiao et al., 2014*). A Benjamini-Hochberg false discovery rate (FDR, Q) of <5% was then used to analyze all the $p$ values (GraphPad Prism 8; GraphPad Software, Inc., San Diego, CA, USA)

**Table 1  Primer sequences and amplicon details.**

| Gene | Accession no. | Primers (Forward and reverse) | Amplicon size (bp) | Efficiency (%) |
|---|---|---|---|---|
| TBP (TATA-box binding protein) | NM_003194.4 | F: CAGTGACCCAGCAGCATCACT<br>R: AGGCCAAGCCCTGAGCGTAA | 205 | 99 |
| Cyclophilin (PPIA, peptidyl-prolyl cis-trans isomerase A) | NM_021130.4 | F: GTCAACCCCACCGTGTTCTTC<br>R: TTTCTGCTGTCTTTGGGACCTTG | 100 | 100 |
| B2M (β-2-microglobulin) | NM_004048.2 | F: TGCTGTCTCCATGTTTGATGTATCT<br>R: TCTCTGCTCCCCACCTCTAAGT | 86 | 98 |
| ACTB (actin beta) | NM_001101.3 | F: GAGCACAGAGCCTCGCCTTT<br>R: TCATCATCCATGGTGAGCTGGC | 70 | 107 |
| 18S rRNA (RNA, 18S ribosomal 5) | NR_003286.2 | F: CTTAGAGGGACAAGTGGCG<br>R: GGACATCTAAGGGCATCACA | 71 | 99 |
| GAPDH (glyceraldehyde-3-phosphate dehydrogenase) | NM_001289746.1 | F: AATCCCATCACCATCTTCCA<br>R: TGGACTCCACGACGTACTCA | 82 | 106 |
| PGC-1α (peroxisome proliferator-activated receptor gamma coactivator 1 α, PPARGC1A) | NM_013261.3 | F: CAGCCTCTTTGCCCAGATCTT<br>R: TCACTGCACCACTTGAGTCCAC | 101 | 104 |
| PGC-1α, isoform 4 | Adapted from (Ruas et al. 2012) | F: TCACACCAAACCCACAGAGA<br>R: CTGGAAGATATGGCACAT | n/a | 114 |
| PPARα (peroxisome proliferator activated receptor α) | NM_001330751.1 | F: GGCAGAAGAGCCGTCTCTACTTA<br>R: TTTGCATGGTTCTGGGTACTGA | 102 | 93 |
| HSPA1A (heat shock protein family A member 1A) | NM_005345.5 | F: GGGCCTTTCCAAGATTGCTG<br>R: TGCAAACACAGGAAATTGAGAACT | 95 | 99 |
| SDHB (succinate dehydrogenase complex iron sulfur subunit B) | NM_003000.2 | F: AAATGTGGCCCCATGGTATTG<br>R: AGAGCCACAGATGCCTTCTCTG | 102 | 104 |
| COX4-1 (cytochrome c oxidase subunit 4I1) | NM_001861.6 | F: GAGCAATTTCCACCTCTGC<br>R: CAGGAGGCCTTCTCCTTCTC | 172 | 104 |
| TFAM (transcription factor A, mitochondrial) | NM_003201.2 | F: CCGAGGTGGTTTTCATCTGT<br>R: GCATCTGGGTTCTGAGCTTT | 110 | 109 |
| CS (citrate synthase) | NM_004077.3 | F: TGGGGTGCTGCTCCAGTATT<br>R: CCAGTACACCCAATGCTCGT | 86 | 111 |
| p53 (tumor protein p53, TP53) | NM_000546.5 | F: GTTCCGAGAGCTGAATGAGG<br>R: TTATGGCGGGAGGTAGACTG | 123 | 102 |
| GLUT4 (solute carrier family 2 member 4, SLC2A4) | NM_001042.2 | F: CTTCATCATTGGCATGGGTTT<br>R: AGGACCGCAAATAGAAGGAAGA | 75 | 104 |
| CPT1A (carnitine palmitoyltransferase 1A) | NM_001876.3 | F: ACAGTCGGTGAGGCCTCTTA<br>R: CCACCAGTCGCTCACGTAAT | 148 | 111 |
| NDUFB3 (NADH:ubiquinone oxidoreductase subunit B3) | NM_002491.3 | F: TCAGATTGCTGTCAGACATGG<br>R: TGGTGTCCCTTCTATCTTCCA | 101 | 109 |
| PDK4 (pyruvate dehydrogenase kinase 4) | NM_002612.3 | F: GCAGCTACTGGACTTTGGTT<br>R: GCGAGTCTCACAGGCAATTC | 84 | 100 |
| VEGFA (vascular endothelial growth factor A) | NM_001025366.3 | F: ACAACAAATGTGAATGCAGACCAA<br>R: CGTTTTTGCCCCTTTCCCTT | 85 | 144 |
| PGC-1β (peroxisome proliferator-activated receptor gamma coactivator 1 β, PPARGC1B) | NM_001172698.2 | F: TCTCGCTGACACGCAGGGT<br>R: GCACCACTGCAGCTCCCC | 130 | 92 |
| NRF1 (nuclear respiratory factor 1) | NM_001293163.2 | F: CTACTCGTGTGGGACAGCAA<br>R: AGCAGACTCCAGGTCTTCCA | 143 | 93 |

| Gene | Accession no. | Primers (Forward and reverse) | Amplicon size (bp) | Efficiency (%) |
|---|---|---|---|---|
| CD36 (CD36 molecule) | NM_001371075.1 | F: ACAGATGCAGCCTCATTTCCA<br>R: TACAGCATAGATTGACCTGCAA | 90 | 119 |
| TFEB (transcription factor EB) | NM_007162.2 | F: CAGATGCCCAACACGCTACC<br>R:GCATCTGTGAGCTCTCGCTT | 140 | 102 |
| UCP3 (uncoupling protein 3) | NM_003356.4 | F: CCACAGCCTTCTACAAGGGATTTA<br>R: ACGAACATCACCACGTTCCA | 70 | 90 |
| UQCRC2 (ubiquinol-cytochrome c reductase core protein 2) | NM_003366.4 | F: GCAGTGACCGTGTGTCAGAA<br>R: AGGGAATAAAATCTCGAGAAAGAGC | 79 | 100 |
| PPARβ/δ (peroxisome proliferator activated receptor β/δ) | NM_006238.4 | F: CATCATTCTGTGTGGAGACCG<br>R: AGAGGTACTGGGCATCAGGG | 125 | 109 |
| PPARγ (peroxisome proliferator activated receptor γ) | NM_138712.3 | F: CTTGTGAAGGATGCAAGGGTT<br>R: GAGACATCCCCACTGCAAGG | 150 | 104 |
| MFN2 (mitofusin 2) | NM_014874.4 | F: CCCCCTTGTCTTTATGCTGATGTT<br>R: TTTTGGGAGAGGTGTTGCTTATTTC | 168 | 134 |

(*Benjamini & Hochberg, 1995*). Lastly, one-way ANOVA with Dunnett test was used to determine the differentially expressed gene targets post-exercise, and significance was defined as adjusted $p$ value <0.05 (GraphPad Prism 8; GraphPad Software, Inc., San Diego, CA, USA). For modelling of the exercise-induced expression pattern of the target genes, the least-squares Gaussian nonlinear regression analysis (curve fitting) was used to model the peak of mRNA expression using GraphPad Prism 8 (GraphPad Software, Inc., San Diego, CA, USA). When examining the relationship between biopsy time associated with peak mRNA expression and the mRNA expression at baseline, basal mRNA expression was calculated from the absolute expression of the target gene at baseline (pre-HIIT at Week 0), determined from $2^{-\Delta Cq}$, multiplied by the dilution of cDNA used in the qPCR reactions. Pearson's correlation coefficient was used to assess the relationship between the basal mRNA expression and the biopsy time with the peak mRNA expression (GraphPad Prism 8; GraphPad Software, Inc., San Diego, CA, USA).

## RESULTS

### Dynamic gene expression response to exercise in human skeletal muscle

The influence of biopsy timing on post-exercise mRNA expression was examined following a single session of HIIE. We measured the mRNA expression of 22 genes (23 isoforms) that have been implicated in the adaptive response to exercise (Table 2 and Table S1). Twelve genes showed significant changes (passed at least one of the statistical tests we applied) at the time that elicited the largest fold-change in mRNA expression, and these times varied from 3 to 72 h. To illustrate the distinct time-course of mRNA expression in response to exercise, and based on the research interest of our group (*i.e.*, mitochondrial and metabolic adaptive responses to exercise), we chose to focus on

**Table 2 Summary of changes in mRNA content following a single session of high-intensity interval exercise (HIIE), measured in nine participants.**

| Gene name | Time-point with highest or lowest fold change | Maximal fold change relative to baseline; Geometric mean (GSD) | 95% CI for fold change | p value | Xiao value | q value | Adj p value |
|---|---|---|---|---|---|---|---|
| PGC-1α | 3 h | 3.2 (1.8) | [2.0–5.0] | 0.0005 | $3.04 \times 10^{-6}$ | 0.0110 | 0.0052 |
| PGC-1α4 | 3 h | 4.5 (2.3) | [2.3–8.5] | 0.0005 | $7.35 \times 10^{-8}$ | 0.0110 | 0.0312 |
| PPARα | 3 h | 4.1 (1.5) | [3.0–5.6] | 0.0003 | $3.88 \times 10^{-7}$ | 0.0110 | 0.0251 |
| CPT1A | 3 h | 2.9 (2.8) | [1.3–6.3] | 0.0503 | 0.0108 | 0.1614 | 0.0159 |
| PDK4 | 9 h | 7.4 (14) | [1.0–56.2] | 0.0315 | 0.0002 | 0.1242 | 0.1944 |
| NRF1 | 24 h | 2.5 (1.5) | [1.8–3.3] | <0.0001 | $5.66 \times 10^{-6}$ | 0.0069 | 0.0072 |
| CD36 | 24 h | 0.1 (2.4) | [0.2–0.4] | 0.0106 | 0.0001 | 0.0563 | 0.1223 |
| TFEB | 24 h | 0.6 (1.5) | [0.4–0.8] | 0.0167 | 0.0500 | 0.0768 | 0.0222 |
| UCP3 | 24 h | 2.4 (4.4) | [0.8–7.4] | 0.0400 | 0.0179 | 0.1380 | 0.3903 |
| p53 | 48 h | 2.3 (1.5) | [1.4–3.8] | 0.0078 | 0.0394 | 0.0454 | 0.1058 |
| PPARγ | 48 h | 1.8 (2.6) | [0.7–1.7] | 0.0147 | 0.0409 | 0.0700 | 0.5395 |
| GLUT4 | 72 h | 0.3 (3.7) | [0.1–1.0] | 0.0004 | $3.15 \times 10^{-7}$ | 0.0110 | 0.0072 |

Note:
The time-point with maximal fold changes, the geometric mean for maximal fold change with geometric standard deviation (GSD), the 95% Confidence Interval (CI), the p value determined by the Mann-Whitney test, the Xiao value determined by a novel posteriori information fusion scheme (*Deshmukh et al., 2021*; *Xiao et al., 2014*), the q value determined by a Benjamini-Hochberg false discovery rate (FDR) of <5%, and the adjusted p value (Adj p value) determined by one-way ANOVA with Dunnett test, are reported for each target gene.

seven genes (eight isoforms) that are related to mitochondrial and metabolic adaptations to training, and which elicited peak expression at different times (Fig. 2, presented in the order of the time-point with the largest fold-change in mRNA expression).

The mean of the mRNA expression of *PGC-1α*, exercise-induced isoform *PGC1α4*, and *PPARα*, increased significantly 3 h post-exercise and was not significantly different from baseline at 9 h post-exercise (Figs. 2A to 2C). All 9 participants had the highest expression level of *PGC1α*, *PGC1α4*, and *PPARα* mRNA at 3 h post-exercise (except 1 participant who had the highest *PGC1α4* mRNA expression at 0 h). A similar result was observed for *CPT1A*; the highest mRNA expression for the group mean, and the highest value for seven out of nine individuals, occurred 3 h post-exercise with values not significantly different from baseline at 9 h post-exercise (Table 2, individual data shown in Tables S2 and S3).

The highest mean for the mRNA expression of *PDK4* occurred at 9 h post-exercise (Fig. 2D). HSP1A1, *SDHB, COX4-1, NDUFB3, VEGFA*, and *PGC1β* showed the greatest, but not significant, induction of mRNA expression at 9 h post-exercise (Table S1).

The mean mRNA expression of *NRF1* was highest at 24 h post-exercise (Fig. 2E), and the mean mRNA expression of CD36 had the greatest decrease at 24 h post-exercise (Fig. 2F). Most participants showed the highest or lowest mRNA expression between 3 and 24 h post-exercise (seven out of nine participants for *NRF1*, and eight out of nine participants for *CD36*, respectively). *TFEB, UCP3, CS, TAFM, UQCRC2*, and *PPARβ/δ* also showed the highest or lowest expression level at 24 h; however, only changes in *TFEB* and *UCP3* reached significance (Table 2 and Table S1).

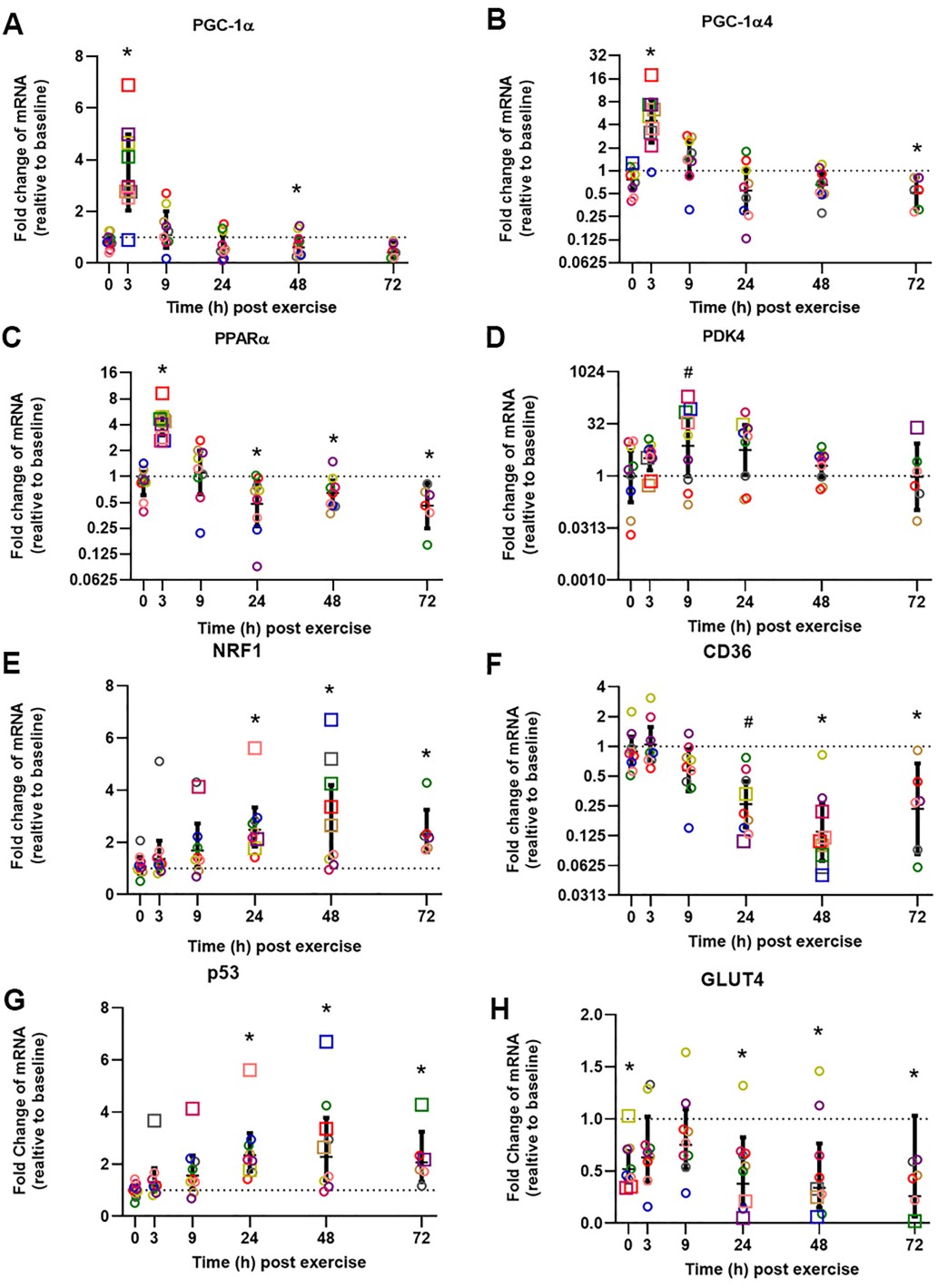

**Figure 2  Relative fold changes compared to baseline for the mRNA content of *PGC-1α* (A), *PGC-1α4* (B), *PPARα* (C), *PDK4* (D), *NRF1* (E), *CD36* (F), *p53* (G), and *GLUT4* (H), following a single session of HIIE.** Muscle biopsies were obtained at rest (baseline) before 4 weeks of high-intensity interval training (HIIT), immediately after the final session of HIIE (0 h), and 3, 9, 24, 48, and 72 h after exercise from nine participants (except for only six participants at 72 h). Symbols (open circles and squares) of the same color indicate mRNA data from one participant; the geometric mean (horizontal bars) ± the 95% confidence interval (CI) are plotted for each graph. The squares indicate the data point with highest or

**Figure 2** (continued)
lowest mRNA content for each participant. A dotted line was used to indicate Y = 1. *Significantly different from baseline, determined by *a posteriori* information fusion scheme and a Benjamini-Hochberg false discovery rate (FDR) of <5%. #Significantly different from baseline, determined by *a posteriori* information fusion scheme only.           

The mean *p53* mRNA expression was significantly greater than baseline from 9 to 72 h post-exercise, with the highest expression 48 h post-exercise (Fig. 2G). *PPARγ* showed significant induction of mRNA expression at 48 h post-exercise (Table 2).

Participants showed a small decrease of *GLUT4* mRNA expression compared to baseline, and the lowest mRNA expression was found at 72 h after exercise (Fig. 2H). The decrease of the mean mRNA expression was significant at all time-points except 3 h and 9 h post-exercise. *MFN2* had the highest mRNA expression at 72 h post-exercise; however, this change was not significant (Table S1).

## Modeling the gene expression response to exercise

We used a least-squares Gaussian nonlinear regression analysis to model the exercise-induced expression pattern of the target genes (Fig. 3). Using the mRNA expression from 0 to 48 h after exercise (the time span in which most of the transcriptional responses were observed), the best curve fit was generated based on the group mean mRNA expression. The predicted peak mRNA expression time was identified based on the regression curve. The modelled peak mRNA expression time ranged from 4.6 h (*PPARα*) to 34.8 h (*p53*) post-exercise (Fig. 4). *CD36, GLUT4, TFEB, and PPARγ* were not used in this analysis as there was no peak in gene expression detected using curve fitting. We then established the time window within which each target gene's expression level was within 90% of its most changed mRNA expression. The genes that responded earlier tended to have a shorter time window within 10% of most changed mRNA expression, whereas the genes that responded later had a larger time window.

## Gene expression timing plotted against basal expression level

To determine if the diverse gene expression timing after exercise depends on its basal expression level in skeletal muscle, we then plotted basal mRNA expression levels of the 12 gene targets with significant changes against the biopsy time that elicited the peak mRNA expression after exercise. There was no clear relationship between the mRNA expression level at baseline and the biopsy time-point that corresponded with the largest change in gene expression ($r = -0.042$, $p = 0.898$, Fig. 5).

## DISCUSSION

The present study examined changes in the mRNA expression of 23 targets at six time-points, over a period of 72 h, after a single session of high-intensity interval exercise in nine healthy participants. We found that changes in gene expression were highly dependent on the biopsy timing, and that the greatest changes in mRNA expression were observed between 3 and 72 h post-exercise. For many genes, there was considerable individual

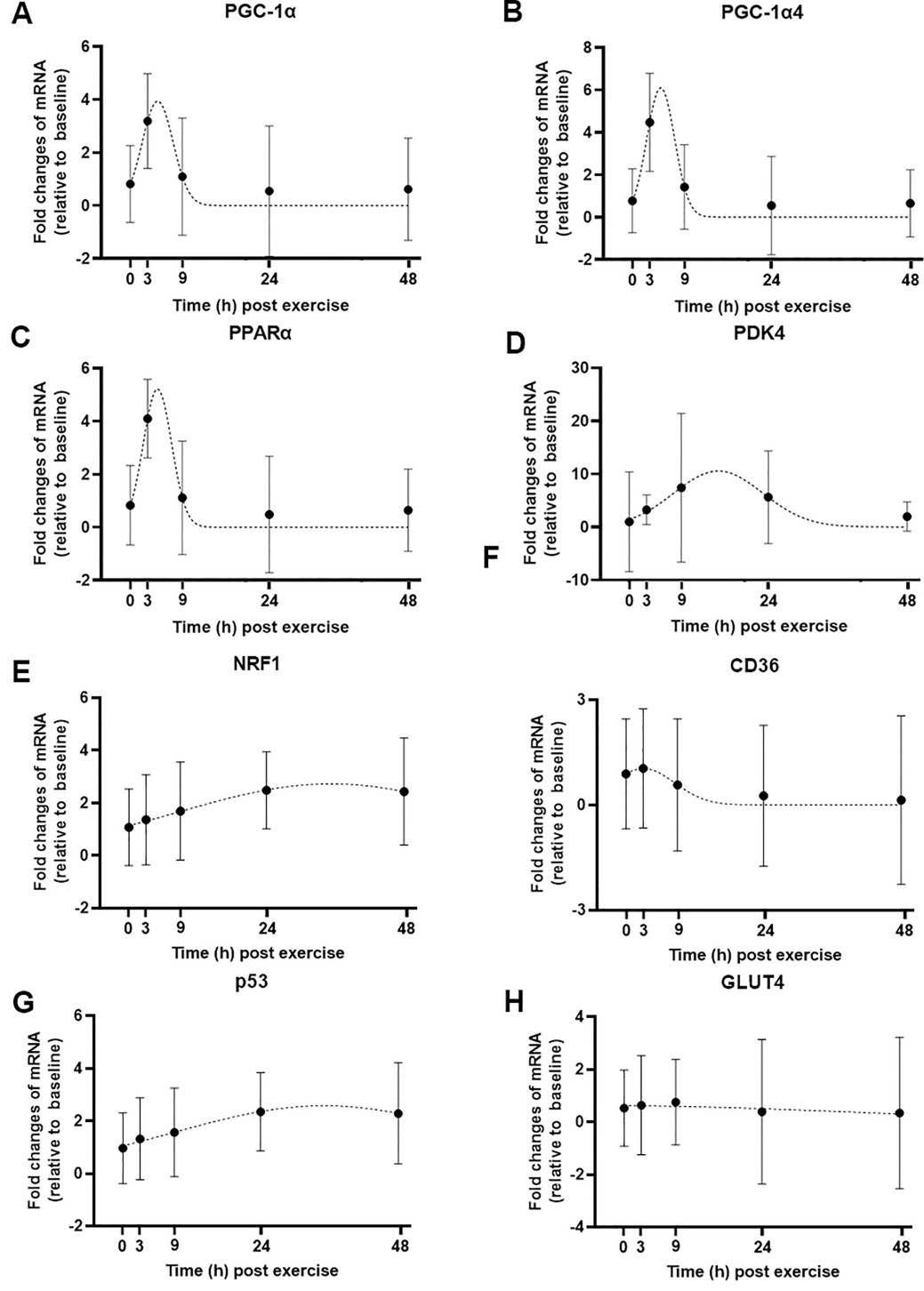

**Figure 3 Curve fitting applied to mRNA changes following a single session of high-intensity interval exercise (HIIE).** Least-squares Gaussian nonlinear regression analysis (dash lines) has been applied to gene expression data for *PGC-1α* (A), *PGC-1α4* (B), *PPARα* (C), *PDK4* (D), *NRF1* (E), *CD36* (F), *p53* (G), and *GLUT4* (H) at five time-points (0, 3, 9, 24 and 48 h following a single session of HIIE) in nine participants. The geometric mean of gene expression is indicated by black dots, error bars are geometric standard deviations.
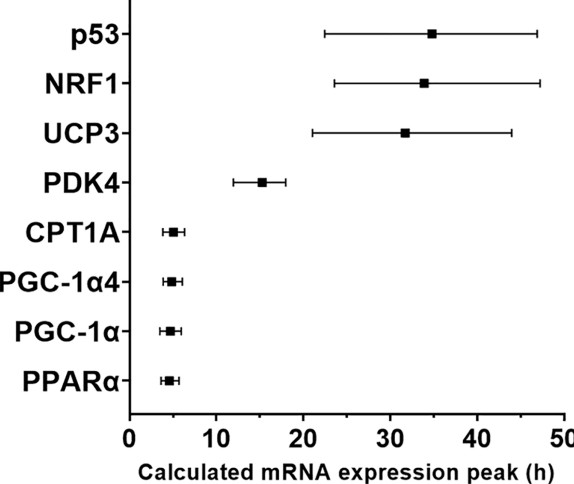

**Figure 4 Modelled time of mRNA expression peak following a single session of high-intensity interval exercise in relation to biopsy timing in nine participants.** The peak expression time (black dots) and the time window for the top 10% of mRNA content (vertical lines) were calculated based on regression analysis and is shown for the seven genes (eight isoforms) for which a peak of gene expression was detected using curve fitting.

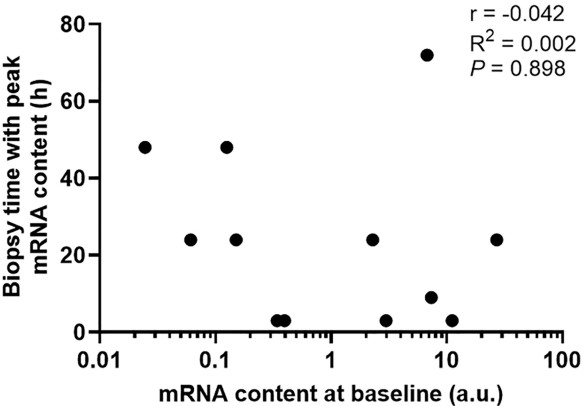

**Figure 5 The biopsy time associated with peak mRNA expression in nine participants plotted against the mRNA content at baseline for 12 gene isoforms.** Pearson's correlation coefficient and *p* value are shown.

variability for the time at which the most changed mRNA expression was observed, especially for genes that responded at later time-points.

We observed distinct temporal patterns of gene expression after exercise, which is consistent with previous research. For example, *Yang et al. (2005)* examined the mRNA expression of several genes at seven time-points from 0 to 24 h after resistance exercise and found that the timing of mRNA induction of their target genes was also variable. They reported that the mRNA expression of *muscle regulatory factor 4* (*MRF4*) and *PDK4* reached their highest levels 4 h post-exercise, whereas *myogenin* and *hexokinase II* (*HKII*) reached their highest levels 8 h post-exercise. Another study assessed exercise-induced gene expression at 3, 48, and 96 h after an endurance exercise session comprised of 60 min
of intense cycling followed immediately by 60 min of intense running (*Neubauer et al., 2014*). Their results showed that the highest mRNA expression of some targets, such as *hemeoxygenase 1* (*HMOX1*) and *integrin beta 2* (*ITGB2*), occurred 96 h post-exercise, whereas other targets, such as *PGC-1α*, had the highest observed mRNA expression at 3 h post-exercise. These findings clearly highlight how biopsy timing has the potential to influence the interpretation of exercise-induced transcriptional responses. For example, if we had only taken a biopsy 3 h post-exercise in the current study (which is common practice), we would have incorrectly concluded that genes such as *PDK4*, *NRF1*, *CD36*, and *p53* were not affected by our exercise stimulus.

The literature is consistent with our observation that the average mRNA expression of *PGC-1α*, as well as the exercise-induced isoform *PGC-1α4*, increased significantly 3 h after a single session of HIIE, and then returned to baseline at 9 h post-exercise. Previous studies have reported that *PGC1α* mRNA increases 2- to 15-fold, 2 to 5 h after a single session of exercise in humans (summarised in (*Granata, Jamnick & Bishop, 2018*)). In another review article, *Islam, Edgett & Gurd (2018)* compared 19 human studies with diverse exercise protocols and muscle sampling time-points; all studies reported an increase in *PGC-1α* mRNA expression, except one study that performed a biopsy 24 h post-exercise. A recent meta-analysis, which integrated 66 published dataset using gene ontology and pathway analyses, confirmed that *PGC-1α* mRNA increases 2.3-fold after a single session of aerobic exercise (*Pillon et al., 2020*).

When we investigated downstream targets of PGC1α protein (a transcriptional coactivator (*Handschin & Spiegelman, 2006*)), we observed that the mRNA expression of *PDK4* was significantly higher at 9 h post-exercise, and the mRNA expression of *NRF1* mRNA reached its highest level at 24 h post-exercise. Our results suggest the downstream targets of the master regulator PGC1α are induced by a single session of exercise but follow a delayed time-course. Our findings support the proposal by Scribbans and colleagues that the absence of a systematic upregulation of PGC-1α targets could be because changes in some targets were not captured by their chosen biopsy time (3 h post-exercise) (*Scribbans et al., 2017*). However, we observed no significant changes in mRNA expression of some downstream targets of PGC-1α, such as *TFAM*, *COX4-1*, and *CS*. A review article by *Islam, Edgett & Gurd (2018)* reported contrasting observations when the expression of those genes targets were examined. For 18 studies that observed an increase in *PGC-1α* mRNA expression, 12 studies reported an increased expression of mitochondrial transcriptional factor *TFAM* with at least one exercise protocol or time-point (biopsy times ranging from 0 to 24 h, but mostly within 6 h post-exercise), and 6 studies reported no change (biopsy times ranging from 0 to 6 h post-exercise). Islam et al. also reported different results for gene expression of *COXIV* (induced at 3 h post-exercise or no change), and *CS* (induced at 5 h post-exercise or no change) from separate studies. There is no clear explanation for the contrasting results; however, the authors suggested this lack of coordination could be due to the divergent temporal expression pattern of different genes and recommended a more thorough investigation of the exercise-induced gene expression time-course in future human studies. The lack of
significant changes in some PGC-1α downstream targets from this present study could be due to the study being underpowered to detect a clear change.

The tumor suppressor p53 is another important regulator of mitochondrial biogenesis and known to be induced by exercise (*Granata et al., 2016*; *Saleem et al., 2011*). We report a 2- to 2.5-fold increase of p53 mRNA from 24 to 48 h post-exercise, with the highest mRNA expression (2.3-fold) observed at 48 h post-exercise. Unlike *PGC-1α*, the exercise-induced changes in *p53* mRNA expression are not consistent across the literature. For example, *Edgett et al. (2013)* reported a small induction of p53 mRNA expression (less than 1.5-fold) after high-intensity interval cycling of varying intensities (73 to 133% $\dot{W}_{peak}$). Conversely, *Hammond et al. (2016)* reported a 2- to 3-fold increase in p53 mRNA expression between 4.5 and 19.5 h after a high-intensity interval running session ($8 \times 5$ min at 85% $\dot{V}O_{2peak}$); their participants also performed a 60-min steady-state run (70% $\dot{V}O_{2peak}$) between the running session and post-exercise biopsies. Differences in the prescribed exercise intensities and volumes may therefore contribute to the inconsistent findings between studies. However, in two of our previous human studies, we observed no significant changes in p53 mRNA expression immediately and 3 h after either moderate-intensity continuous (63% $\dot{W}_{peak}$) or "all-out" sprint interval cycling sessions (*Granata et al., 2017*), nor after high-intensity interval cycling (~79% $\dot{W}_{peak}$) whether preceded by a prior exercise session or not (*Andrade-Souza et al., 2020*).

The half-life of mRNA is important for the kinetics of gene expression, as mRNA expression is determined by the rates of both RNA synthesis and degradation. It has been reported that many transcription factors and regulatory proteins have short half-lives (*Uhlitz et al., 2017*; *Yang et al., 2003*). In a study that analyzed mRNA half-lives in human B-cells, the authors reported that genes involved in transcription factor activity, transcription, and transcription factor binding, are short-lived, with median half-lives ranging between 3.6 to 4.3 h, whereas genes involved in glucose and fatty acid metabolic process have longer median half-lives of 7.5 to 10.1 h (*Friedel et al., 2009*). This matches our observation that transcription factors and coactivators, such as *PGC-1α* and *PPARα*, are fast-responding genes following exercise. In contrast, genes with functions in glucose metabolism (*PDK4*) and fatty-acid metabolism (*CD-36* and *UCP3*) took longer to be induced after our exercise stimulus. It has also been suggested that the kinetics of mRNA induction are influenced by basal expression levels, in addition to the activation of transcription and mRNA turnover (*Booth & Neufer, 2012*). However, our results indicate that the diverse expression timing of different genes after a single session of exercise stimuli depends more on the function/role of the target gene in the process of the adaptative response than its basal expression level in skeletal muscle (*i.e.*, time to peak expression did not seem to be related to mRNA expression level at baseline; Fig. 5).

Due to limitations regarding the number of muscle biopsies that can be obtained in a single experiment, it was not possible to determine the precise peak expression time of each target gene in the present study. Thus, we decided to model the peak expression timing of target genes based on the observed mRNA changes in all participants. Previous time-course studies in human subjects (*Mathai et al., 2008*; *McGinley & Bishop, 2016a*; *Pilegaard, Saltin & Neufer, 2003*; *Yang et al., 2005*), as well as our present experiment,
observed that the majority of the exercise responsive genes followed a similar pattern: an initial upregulation to an observed peak level followed by a return to baseline levels. Based on these observations, we chose least-squares Gaussian nonlinear regression modelling to analyze the expression pattern using the gene expression data from 0 to 48 h post-exercise. The reasons for only including the time-points within the first 48 h post-exercise include: (1) the largest changes in mRNA expression for the majority of genes occurred within the first 2 days after exercise; (2) the low number of participants available for the 72-h post-exercise biopsy. Based on this model, we mapped out the time window eliciting the highest or lowest 10% of gene expression after exercise (Fig. 4). One limitation of our modelling method, however, is that this parametric regression is strongly based on the assumption that the genes followed the modelled pattern. Furthermore, the large individual variation between participants, the relatively low number of participants, and the limited number of muscle biopsies taken, undoubtedly affected the accuracy of our regression analysis. An even more comprehensive study, with more participants, more biopsy time-points, and a greater number of genes (assessed with RNAseq) is necessary to construct a more accurate picture of the kinetics of mRNA expression in response to exercise.

To standardize the exercise dose, we prescribed the same relative intensity and duration of HIIE for all participants. Nevertheless, we still observed large variability in both the timing and the magnitude of the transcriptional response between individuals. This is consistent with previous research showing that not all individuals respond the same way to a standardized exercise dose (*Bouchard & Rankinen, 2001*; *Jamnick et al., 2020*; *Ross, de Lannoy & Stotz, 2015*). This individual response could be due to genetic background, non-genetic biological and behavioral factors, circadian fluctuations, and technical or biological variability associated with the sampling of human skeletal muscle and the analysis of exercise-induced mRNA (*Islam et al., 2019*; *Ross et al., 2019*). We add that the timing of the observed peak for exercise-induced expression of different genes also differed between individuals. This further highlights that it is probably not possible to choose a single post-exercise biopsy time-point that will capture changes in the expression of specific genes for all individuals.

In the present study, participants underwent 4-weeks of HIIT before the single session of HIIE and post-exercise muscle sampling. This prior training period may have influenced our interpretation of results when compared to the pre-training baseline values. However, *Miyamoto-Mikami et al. (2018)* reported that of 24,838 genes assessed, there were only 152 differentially expressed genes after 6 weeks of exercise training: 79 genes significantly up-regulated (fold change between 1.2 to 2) and 73 genes significantly down-regulated (fold change between 0.5 to 0.8). Only 1 of our target genes (*PGC-1α*) overlapped with their 152 differently expressed genes. This suggests the fold change in gene expression changes detected in this present study are likely to be representative of the adaptive response to the single session of exercise, rather than the training. Another study has also shown there was no difference in the mRNA expression of *PGC-1α*, *PGC-1β*, *TFAM*, *CS* and *COXIV* after 10 days of intensive cycling training (*Stepto et al., 2012*). It is still possible that the magnitude or lack of fold change in the

expression of some gene targets could be due to the training period in our study. For example, it has been shown that training was able to increase *GLUT4* mRNA expression (*Stuart et al., 2010*), which could explain the lack of induction of *GLUT4* mRNA expression following exercise that we observed in trained individuals. Despite this limitation in our study design, it is important to note that our primary interest was to investigate temporal changes in gene expression and not the fold changes from baseline or the expression of every individual gene target.

Another limitation of the present study is the possible influence on mRNA expression from non-exercise factors, such as repeated muscle biopsies, the post-exercise nutrition control, and circadian rhythms. It has previously been demonstrated that the expression of some gene targets was induced by stimuli other than exercise, such as feeding and the stress from repeated muscle biopsies. However, this observation could be due to the high-glycemic meal that was provided to the participants after the first biopsy (*Vissing, Andersen & Schjerling, 2005*). Other studies have clearly demonstrated that prior muscle biopsies had no effect on mRNA expression of specific genes, including *PGC-1α*, *PDK4*, and *GLUT4* (*Lundby et al., 2005*; *Psilander, Damsgaard & Pilegaard, 2003*). Furthermore, other studies have shown that some mRNA transcripts follow a circadian pattern of expression (*Kemler, Wolff & Esser, 2020*; *Saner et al., 2020*); however, considering that participants all performed the exercise session and ate meals at the same time of the day this is unlikely to have confounded the results of this study.

Another critical factor to interpret the gene expression data obtained from human exercise studies is the use of the most appropriate statistical analyses. The common issues with human exercise studies include relatively small sample sizes, large individual variability, and not correctly controlling for multiple comparisons. Future research investigating a time-course analysis of exercise-induced transcriptional responses in humans should employ larger sample sizes and ensure that statistical analyses adequately control for multiple comparisons.

It is important to note that our results are likely to be specific to the participants recruited and the exercise employed. It has previously been reported that both training and fitness level can affect exercise-induced gene expression (*Popov et al., 2018*; *Popov et al., 2017*). Thus, the time courses we have reported cannot be assumed for other populations (*e.g.*, sedentary and untrained participants, or elite athletes). Many studies have also reported intensity-dependent changes in gene expression (*Edgett et al., 2013*; *Egan et al., 2010*; *Sriwijitkamol et al., 2007*). It remains to be determined if the time courses we have reported are also influenced by exercise intensity.

It is also worth mentioning that adaptive responses to exercise training consist of changes in steady-state protein abundance, along with subsequent functional adjustments (*e.g.*, changes in enzyme activities) (*Perry et al., 2010*). These adaptations are collectively determined by rates of mRNA transcription, mRNA degradation, translation, and protein degradation. Nonetheless, it has been reported by *Li, Bickel & Biggin (2014)* that variance in mRNA expression explains more than 80% of the variance in protein levels. This suggests that better characterizing transcriptional responses to exercise does contribute to understanding the adaptive response to exercise training.

## CONCLUSIONS

In summary, this study monitored the temporal expression of mRNA with multiple time-points over 72 h after a single exercise session. We observed distinct temporal patterns for the expression of different genes, with the time for the highest observed gene expression varying from 3 to 48 h post-exercise. These findings highlight an important limitation when studying the molecular responses to exercise, where few (2 to 3) biopsies are commonly sampled in a short time frame (*i.e.*, less than 24 h post exercise) to examine transcriptional responses to exercise. These results further emphasize the importance of carefully planning biopsy time-points to best capture, and interpret, exercise-induced changes in genes of interest.

## ACKNOWLEDGEMENTS

We are very grateful to the participants for their time and effort. We also thank Dr Mitch Anderson for performing some of the muscle biopsies.

### Funding

This study was supported by an Australian Research Council Grant DP140104165 (to David J Bishop). The funders had no role in study design, data collection and analysis, decision to publish, or preparation of the manuscript.

### Grant Disclosures

The following grant information was disclosed by the authors:
Australian Research Council: DP140104165.

### Competing Interests

The authors declare that they have no competing interests.

### Author Contributions

- Jujiao Kuang conceived and designed the experiments, performed the experiments, analyzed the data, prepared figures and/or tables, authored or reviewed drafts of the paper, and approved the final draft.
- Cian McGinley conceived and designed the experiments, performed the experiments, analyzed the data, authored or reviewed drafts of the paper, and approved the final draft.
- Matthew J-C. Lee performed the experiments, analyzed the data, prepared figures and/or tables, and approved the final draft.
- Nicholas J. Saner performed the experiments, analyzed the data, authored or reviewed drafts of the paper, and approved the final draft.
- Andrew Garnham performed the experiments, authored or reviewed drafts of the paper, and approved the final draft.
- David J. Bishop conceived and designed the experiments, analyzed the data, authored or reviewed drafts of the paper, and approved the final draft.

## Human Ethics

The following information was supplied relating to ethical approvals (*i.e.*, approving body and any reference numbers):

All procedures were approved by the Victoria University Human Research Ethics Committee.

## Data Availability

The raw measurements are available in the Supplemental Files.

## Supplemental Information

Supplemental information for this article can be found online at http://dx.doi.org/10.7717/peerj.12856#supplemental-information.

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
