# Peer review of "Interpretation of exercise-induced changes in human skeletal muscle mRNA expression depends on the timing of the post-exercise biopsies"

_PeerJ, doi:10.7717/peerj.12856_

## Round 0.1 · original submission · Major Revisions

The authors should address the concerns raised by the reviewers, who have provided detailed and constructive analysis of the manuscript.

Reviewer 1 ·

Basic reporting

• The subheading “Conclusions” needs to be added to the Abstract
• Lines 59-82: Obviously there are many, many studies with measurement of post-exercise mRNA expression and a good cluster of them are referenced here, but to imply that there is a dearth of those that go beyond the first 24 hours, or that few have more than 2 or 3 timepoints is not a fair representation of the literature in my opinion. For example, the following (not exhaustive list) could be said to be exceptions to the points you make, granted some are resistance exercise.
o Pilegaard et al 2005 https://pubmed.ncbi.nlm.nih.gov/16092055/
o Schmutz et al 2006 https://pubmed.ncbi.nlm.nih.gov/16362354/
o Louis et al 2007 https://pubmed.ncbi.nlm.nih.gov/17823296/
o Klossner et al 2007 https://pubmed.ncbi.nlm.nih.gov/17701424/
o McKay et al 2008 https://pubmed.ncbi.nlm.nih.gov/18818249/
o Deldicque et al 2008 https://pubmed.ncbi.nlm.nih.gov/18048590/
o Murton et al 2014 https://pubmed.ncbi.nlm.nih.gov/24265280/
o Jensen et al 2015 https://pubmed.ncbi.nlm.nih.gov/25677542/
o Gidlund et al 2015 https://pubmed.ncbi.nlm.nih.gov/26089547/
o Ogborn et al 2015 https://pubmed.ncbi.nlm.nih.gov/25695287/
o Annabilini et al 2019 https://pubmed.ncbi.nlm.nih.gov/31143128/
My point is not made to speak against the value of your work, but I would suggest reframing the introduction to acknowledge there have been many studies of this nature and this is yet another but has a slightly different focus. One of the benefits of PeerJ is that the novelty is not a consideration here. Depending on your view of these references, a change to line 366 may be warranted.
• Line 369: “We found that the changes in gene expression were highly dependent on the biopsy timing”. I would say that you need to say “We found that the conclusions/assertions/interpretations about the changes in gene expression…”. Just check for instances like this throughout but generally my sense is that you have may this qualification most of the time.
• Line 387-390: Excellent, this is the central issue in the manuscript and in the field so I suggest making much more of this point in the Introduction and Discussion. Erroneous interpretations plague the field.
• Line 396: proposed by whom? If so, why not measure it here? Overall I found this section and the IEG narrative to not work. I think it is very worthwhile to discuss the PGC-1 data but not as the current narrative. On line 411, the implication seems to be that the IEGs must be translated before changes in other mRNAs take place and that this is all very sequential. I find that hard to believe. A better explanation is that (as you describe later), the half lives of regulation of specific genes (independent of the IEG response) probably explains the gene-specific time courses. My read of your Discussion is that there is a lot of over-interpretation of what a change in mRNA means because it reads as though you are proposing that a change in mRNA of gene A indicates the role the function of the that gene’s protein A will have on gene B. That is simply not the case. So lines 391-453 need some work in my opinion as it ends up focussing too much on gene regulation of different genes (for which you do not have data on regulatory elements) rather than the interpretation of the data you have to hand in the context of your overall aims.

Experimental design

• Overall the methods are described in nice detail, and experimental design has a lot of strong control/standardisation including the dietary control and the timing of later biopsies being devoid of influence of morning nutrition and circadian variation. In my opinion, the design with the baseline being 4 weeks before the HIIE is a major weakness, although you have done as good a job as you can in justifying it and cross-checking expression data in the straining study you reference. However, whatever about the basal expression of mRNA, the HIIE is performed by a muscle that is fundamentally different to baseline and given the effect of a training intervention on the acute molecular response to exercise, I think this point should at least be acknowledged in the Discussion. e.g. the lack of acute change or modest change in some genes may be a training effect. For example, the lack of change in GLUT4 (if I am interpreting the data correctly) could be explained by this point.
• Related to this point, on line 125 you refer to the baseline biopsy timing being in order to reduce damage/stress but this seems to be a moot point given that there are then multiple biopsies in close temporal proximity and only 1 cm apart (which in my mind is unconventionally close together)?
• Could you add a statement about the time that the HIIE took place after the most recent HIIE session?
• Line 159: a sentence or two to explain the calculation of Wpeak is more preferable than the current statement.
• I suggest that adding a horizontal dotted line a y=1 to indicate baseline expression would be a useful addition to all graphs.
• It might help the reader to also have the graphs in Figs 2 and 3 organised in terms of regulators as one cluster e.g. top half, and the mitochondrial and metabolic markers as another cluster e.g. bottom half. At present, I don’t see a pattern to the way that the graphs have been organised. Just a tentative suggestion.

Validity of the findings

• My major concern is the statistical analysis and given much of the rest of the manuscript hinges on these findings, I would expect that there may be major revisions of this submission depending on the outcomes of my concerns.
o Have the data been tested for normality in order to proceed with t-tests?
o What is the justification for using a 10%FDR rather than 5% which would be more consistent with the widespread use of p<0.05. Based on the q-values stated in Table 2, at an FDR of 5%, only 5 genes would have a peak that is statistically significant. I presume that the using the BH correction is in anticipation that the something like a Bonferroni is to stringent? Thus in both instances, you are increasing the chances of a Type I error. Of course this can be acceptable as an approach but in my opinion needs to be justified (small sample size, noisy data set, exploratory analysis, etc) with appropriate scientific references.
o That said, given the research design and the stated aims, in my opinion a one-way ANOVA on the raw data (not fold change) with Dunnett’s post-hoc correction versus baseline is the appropriate statistical test.
• The major issue with all of the above is that you are arguing throughout the submission that the timing of biopsy central to interpretation of the response to acute exercise, yet one could also argue that the statistical approach is also a major factor.
• In Table 2, I would then ask the question is there such a thing as time to peak and magnitude of peak if there is not really a statistically significant peak in expression? This is particularly relevant to line 301-304, which are discussing changes where there are some gene with no changes evident based on the statistical analysis; and whether the data in Fig 4 and 5 can be considered valid if there were not statistically significant peaks?
• Given the stated aim is around time course and how that affects interpretation, I find it conspicuous that you then ignore something like NRF-1 at 24 h given its statistical significance and especially given NRF-1 is a mitochondrial regulator, which you state is your interest (GLUT4 is not mitochondrial for example). Similarly if I am interpreting the data correctly, you have a major downregulation of CD36, so would it not be of interest to discuss a downregulated gene give the premise of the submission about the importance of interpretation?

Additional comments

• Line 62 – Egan 2016 is not an appropriate reference here. Egan 2013 PLoS ONE perhaps?
• Fig. 2 legend: Typo in title after GLUT4? And “except 6 participants”, there is no mention of the other time points being n=9.
• I question the value of the 72 h data given (i) the n-size of 6 at 72 h, (ii) and the practical relevance of gene expression change at 72 h post-exercise considering that in both health and performance domains, the next training session will have almost certainly have taken place before this time.

Reviewer 2 ·

Basic reporting

The manuscript is well written and very clear. The authors clearly state the purpose of the work and why the study is relevant with the current gap in knowledge surrounding changes in mRNA expression following endurance type exercise over an extended 'acute' time course. Although a lot of literature exists surrounding acute mRNA responses to exercise the authors have well referenced relevant literature. The discussion although well written is largely focused on PGC1-a, especially the first half, I believe it can be shorter and more concise.

Experimental design

Ln 112 – Is the reported VO2 peak value from baseline or following the HIIT intervention?
Ln 117 – Please confirm how many days the familiarisation trial was completed prior to the experimental trial for the collection of the ‘pre’ sample
Ln 119-121 – Please confirm how many days after the GXT the ‘pre’ biopsy would have been taken

Although the research question is well defined I do have several concerns regarding the design and timing of muscle sampling. Can the authors please address the following:

The timing of the ‘pre’ biopsy, 4 weeks prior to the acute post-exercise biopsies is a concern. For example the work by Miyamoto-Mikami et al. 2018, which the authors reference, did show that PPARGC1A (PGC1a) expression was up-regulated after 4 wks of HIIT, other groups have also demonstrated this (PMID: 23285255, PMID: 27604398). Other mitochondrial-related genes (PMID: 23285255), GLUT-4 (PMID 20010125) may be affected by relatively short training programs.
Although the purpose of the current study is to better describe the acute affect of exercise on gene expression an increase in basal expression of gene expression following training (i.e. if the ‘pre’ biopsy was taken after the training intervention) may not affect the interpretation of results. I do think the authors should acknowledge this as a limitation.

As the muscle biopsies following HIIE were completed in a trained state (following 4 weeks of HIIT) the authors should emphasize that the time points identified which correspond to the greatest increases in gene expression are in trained individuals. Training can alter the response to acute exercise bouts and therefore the timing in which the greatest increase was observed in these individuals may not be the same in untrained individuals.

Can you clarify how you identified the time point that elicited ‘the most extreme mRNA content of each gene’ was it simply the greatest numerical fold change? Other than the time points presented in table 2 were other time points also significantly different from pre? A few are mentioned in the text but if there are others this should be included in the table also and if this isn’t the case it can be stated that changes were only evident at the time point presented in the table.

Ln 264 - Can you please clarify how basal mRNA content was calculated?

Ln 355 – as the authors state a clear relationship isn’t observed between mRNA expression level at baseline and the biopsy time point that had the biggest change in gene expression, this is clear in figure 5. But can you confirm how the lack of a relationship was determined? Was a correlation run? Why are only 5 genes highlighted in the figure 5?

Validity of the findings

The authors aimed to address a gap in knowledge with respect to temporal responses in gene expression following a bout of exercise. As mentioned above I think it is important to highlight that these results are in trained individuals (completed 4 wks of HIIT) and the increases reported may not occur at the same time point in untrained individuals.

The authors use a relatively small number of participants n=9 and n=6 at 72h, is this small sample size sufficient to detect differences in gene expression from baseline especially based on the large variability in responses?

Can the results described be extended to other types of exercise, specifically different modes of endurance exercise? This could be better reflected in the conclusion highlighting for the reader that different types of acute exercises may result in increased gene expression at different time points.

Additional comments

This is likely an oversight, however can you please confirm that the raw C(q) scores that have been submitted are actually the mean of the replicates?

---

## Round 0.2 · Major Revisions

Although both reviewers agree that the manuscript has been greatly improved, a further moderate revision has to be made, especially in relation to the remaining concerns of reviewer 2.

Reviewer 1 ·

Basic reporting

Nothing further to add from the first round of review.

Experimental design

Nothing further to add from the first round of review.

Validity of the findings

Nothing further to add from the first round of review.

Additional comments

Thanks for your revisions and your efforts to take my suggestions onboard as well as addressing my main concerns. I do have much more confidence in the results given the robust update to the analysis and therefore the manuscript is much improved. I only have very minor comments/typos that remain that you can take or leave as you wish:

- It may be worth removing Annibalini et al. 2019 despite my suggestion to include it as the timecourse there is a mix of muscle and blood, for your point about the timecouse of change in muscle, there are only two data points so it probably doesn’t support the point. That was my mistake to include that in the list.

- Throughout the manuscript, the terms mRNA content and mRNA expression are used (mRNA level/ expression level is also used in some places) - is there a specific reason why one is used in one instance and not in another? If not, it may be better to be consistent with one or the other?

- The phrase “most extreme mRNA” does not sit well with me. I know this is a change from the previous version but I am not sure it improves things. Would something like most changed/greatest change/peak/nadir etc work better?

- Line 364 - "exam" should be examining I think.

- Line 653 - “when the expression of downstream those genes targeted by PGC-1α was targets were examined” does not make sense to me and may need refining.

- Lines 695-699 - could the differences between exercise challenges have a role to play here? Even if not, it may be useful to add some description to what was used in each study.

Reviewer 2 ·

Basic reporting

The majority of my comments/concerns have been addressed. The introduction and discussion are improved and reference a greater number of previous studies to better describe the current state of knowledge in the field.

For clarity can you please describe the difference between figures 3 and 4? Do both figures use regression analysis to demonstrate the predicted peak in mRNA expression between 0-48h post HIIE? If this is the case there is no need to replicate the results in both figures (PGC-1a, PGC-1a4, PPARa, p53, PDK4, NRF1).

Experimental design

Thank you for addressing all my initial comments related to methods.
However, with the inclusion of new statistical tests can you please clarify your approach?
Did you use the Mann-Whitney U test to produce the p-value to enter into the Xiao “fusion scheme”? I do appreciate that an FDR adjustment was used on the Xiao values. I am a bit confused as to why the one-way ANOVA was performed, could you not run an ANOVA first (rather than the Mann-Whitney) then use the posteriori Xiao fusion scheme as a posthoc analysis?

Validity of the findings

Ln 406: Gene expression timing plotted against basal expression level
Can you be clearer on the relevance of this analysis? I appreciate the correlation shows no relationship between basal levels of gene expression and time of ‘peak’ expression but I don’t think the physiological relevance is well discussed.
Would evaluation of each gene individually not be more valuable? For example, are basal levels of a gene of interest (i.e. PGC1-a) related to time of increase in each participant? Especially if the authors state “For many genes, there was considerable individual variability for the time-point at which the most extreme mRNA content was observed, especially for genes that responded at later time-points.” (ln 424-426)

Additional comments

As a general comment when discussing previous work describing the impact of exercise on gene expression I think it would be useful to include some information surrounding the intervention used. Occasionally this information is included but not always for example Ln 514-516 Hammond et al 2016 was following a steady state bout of exercise (which was preceded by a HIIT session several hours earlier) whereas Edgett et al 2013 used a bout of HIIE.
Ln 301: when examining
Ln 548-549: consider revising for clarity
Ln 552: “Based on this model, we mapped out the time window eliciting the highest or lowest 10% of gene expression after exercise (Fig. 3).” is this not Figure 4?

---

## Round 0.3 · accepted · Accept

All the issues raised by the reviewers have been exhaustively addressed by the authors.

Reviewer 1 ·

Basic reporting

No further comments.

Experimental design

No further comments.

Validity of the findings

No further comments.

Additional comments

Congratulations on a nice piece of work.

Reviewer 2 ·

Basic reporting

no comment

Experimental design

no comment

Validity of the findings

no comment

Additional comments

Thank you for addressing all my revisions I don't have any further comments.